# CRISPR/Cas9-Mediated Disruption of the *lef8* and *lef9* to Inhibit Nucleopolyhedrovirus Replication in Silkworms

**DOI:** 10.3390/v14061119

**Published:** 2022-05-24

**Authors:** Yujia Liu, Xiaoqian Zhang, Dongbin Chen, Dehong Yang, Chenxu Zhu, Linmeng Tang, Xu Yang, Yaohui Wang, Xingyu Luo, Manli Wang, Yongping Huang, Zhihong Hu, Zulian Liu

**Affiliations:** 1Key Laboratory of Insect Developmental and Evolutionary Biology, Center for Excellence in Molecular Plant Sciences, Shanghai Institute of Plant Physiology and Ecology, Chinese Academy of Sciences, Shanghai 200032, China; liuyujia@cemps.ac.cn (Y.L.); zhangxiaoqian7@aliyun.com (X.Z.); chendongbin@stu.syau.edu.cn (D.C.); yangdehong@cemps.ac.cn (D.Y.); m17853512736@163.com (C.Z.); lmtang2016@cemps.ac.cn (L.T.); yangxu@cemps.ac.cn (X.Y.); yhwang@cemps.ac.cn (Y.W.); xy384@outlook.com (X.L.); yphuang@sibs.ac.cn (Y.H.); 2University of Chinese Academy of Sciences, Beijing 100049, China; 3State Key Laboratory of Virology, Wuhan Institute of Virology, Chinese Academy of Sciences, Wuhan 430071, China; wangml@wh.iov.cn (M.W.); huzh@wh.iov.cn (Z.H.)

**Keywords:** BmNPV, CRISPR/Cas9, transgenic silkworm, antiviral therapy

## Abstract

*Bombyx mori* nucleopolyhedrovirus (BmNPV) is a pathogen that causes severe disease in silkworms. In a previous study, we demonstrated that by using the CRISPR/Cas9 system to disrupt the BmNPV *ie-1* and *me53* genes, transgenic silkworms showed resistance to BmNPV infection. Here, we used the same strategy to simultaneously target *lef8* and *lef9*, which are essential for BmNPV replication. A PCR assay confirmed that double-stranded breaks were induced in viral DNA at targeted sequences in BmNPV-infected transgenic silkworms that expressed small guide RNAs (sgRNAs) and Cas9. Bioassays and qPCR showed that replication of BmNPV and mortality were significantly reduced in the transgenic silkworms in comparison with the control groups. Microscopy showed degradation of midgut cells in the BmNPV-infected wild type silkworms, but not in the transgenic silkworms. These results demonstrated that transgenic silkworms using the CRISPR/Cas9 system to disrupt BmNPV *lef8* and *lef9* genes could successfully prevent BmNPV infection. Our research not only provides more alternative targets for the CRISPR antiviral system, but also aims to provide new ideas for the application of virus infection research and the control of insect pests.

## 1. Introduction

Sericulture is one of the main sources of income for farmers in China, India, Brazil, Vietnam, and Thailand [1,2]. Silkworms are susceptible to a variety of diseases caused by bacteria, fungi, protozoans, and viruses that can impair cocoon production. This results in considerable economic loss. *Bombxy mori* nucleopolyhedrovirus (BmNPV) is one of the viral pathogens that is a threat to silkworm growth and to the sustainability of the sericulture industry in China and worldwide.

BmNPV is a member of the genus *Alphabaculovirus* of the family *Baculoviridae*, which are double-strand DNA viruses with large circular genomes. The BmNPV genome is about 128 kilobase pairs [3,4] with a structure very similar to that of the Autographa californica multiple nucleopolyhedrovirus (AcMNPV) genome [4,5,6]. BmNPV contains 38 core genes which are present in all sequenced baculovirus genomes. The genes of baculoviruses are transcribed in four temporally regulated expression phases during the infection cycle: immediate early, delayed early, late, and very late phases [4]. The expression of immediate early genes completely depends on the gene expression products of host cells without any virus gene products. Delayed early genes rely on the expression products of immediate early genes to synthesize proteins necessary for viral DNA replication. The late and very late genes were changed to use the RNA polymerase encoded by the virus in AcMNPV [7,8,9]. The baculovirus RNA polymerase from AcMNPV has four subunits encoded by *p47*, *lef4*, *lef8*, and *lef9* [10]. Their homologs have been identified in BmNPV [11]. The *lef8* and *lef9* genes are indispensable for viral replication, and LEF8 and LEF9 contain conserved motifs present in the large subunits of other DNA-directed RNA polymerases [12,13]. The *lef8* is located at positions 35,594–38,227 in the BmNPV genome; it has a length of 2634 bp. The *lef8* is an essential gene, since *lef8*-KO BmNPV mutants are unable to initiate very late transcription or produce infectious virions [14,15]. The *lef9* is located at positions 44,402–45,874 in the BmNPV genome; it has a length of 1473 bp. The gene encodes a protein of 491 amino acids with a predicted molecular mass of 54 kDa [16]. Although *lef8* and *lef9* encode homologs of RNA polymerase subunits, the functions of these genes in BmNPV have not been characterized.

Various methods have been used to downregulate pathogen genes in silkworms, including RNA interference-mediated gene silencing [17,18] and the overexpression of endogenous or exogenous resistance genes [19,20,21]. In a previous study, we engineered silkworms that expressed small guide RNAs (sgRNAs) and Cas9 to direct the disruption of *ie-1* and *me53* intermediate early genes and showed cleavage of the BmNPV genome DNA to promote virus clearance [22]. Recently, we developed an inducible CRISPR/Cas9 antiviral system targeting BmNPV, and the results showed enhanced antiviral efficacy [23].

In order to determine the efficiency of multigene editing and to identify more efficient antiviral target sites in this study, we targeted the *lef8* and *lef9* genes. The *lef8* and *lef9* transgenic hybrid lines also had significantly higher survival rates upon inoculation with 10^5^ and 10^6^ BmNPV occlusion bodies (OBs) per larva. Thus, gene editing of *lef8* and *lef9* increases the survival of silkworm larvae infected with BmNPV, suggesting a route towards controlling the viral infection of silkworms. Our research provides more alternative targets for the CRISPR antiviral system, as well as new insights into the application of virus infection research.

## 2. Materials and Methods

### 2.1. Silkworm and Virus Inoculation

The multivoltine, nondiapausing silkworm strain *B. mori* (Nistari) [24] was used in this study. Larvae were reared on fresh mulberry leaves at 25 °C with 70–85% relative humidity. To propagate BmNPVs (GenBank accession no. JQ991008), the occlusion bodies (OBs) were harvested from the infected larvae. The virus stock was prepared as previously described [22,25].

### 2.2. Vector Construction

The *piggyBac*-based transgenic plasmids were constructed for target gene editing. The sgRNA targets in *lef8* and *lef9* were selected according to the GG-N^19^-GG rule [26]. Plasmids were constructed as described previously [22]. The plasmid *pBac* [*IE1*-EGFP-*IE1*-Cas9] (*IE1*-Cas9) was used for the constitutive expression of Cas9 and EGFP under control of the *IE1* promoter. The four targeting cassettes expressing the gRNAs targeting *lef8* and *lef9* under the separate control of the silkworm small nuclear RNA promoter *U6* were constructed through a series of cloning steps using the ClonExpress MultiS One Step Cloning Kit (Vazyme) to generate the final plasmid *pBac* [*IE1*-EGFP-*IE1*-Cas9-U6-*lef8* + *lef9* sgRNAs] (Figure 1A). Primers used in this study are listed in Appendix A.

### 2.3. Generating Transgenic Silkworms

The *piggyBac* helper plasmids and transgenic plasmids were microinjected into G_0_ preblastodermal embryos for germline transformation. The putative transgenic adults were mated with wild type moths, and the G_1_ mutants were scored for ubiquitous presence with the EGFP fluorescence using fluorescence microscopy (Nikon AZ100). Three healthy transgenic lines (TG-A, TG-B, and TG-C) were obtained and we used them for follow-up experiments.

### 2.4. Inverse Transcription-PCR and Sequencing of the Transgenic Silkworm

Inverse PCR was performed as previously described [27] to determine the insertion locations in the transgenic silkworms. Briefly, genomic DNA was extracted from G_1_ transgenic silkworms by standard SDS lysis–phenol treatment after incubation with proteinase K, followed by RNase treatment and purification. DNA was digested with *Bfuc**I* and circularized by ligation overnight at 37 °C. PCR amplification was carried out using the circularized fragments as templates with primers designed from the 3′ arm of the *piggyBac* vector. All primers are listed in Appendix A.

### 2.5. Viral Inoculation and Mortality Analyses

Newly exuviated third instar transgenic silkworms from lines TG-A, TG-B, and TG-C, in addition to WT silkworms (72 per group) were exposed to BmNPV. The silkworms, which were starved for 12 h, were inoculated orally with a suspension of 10^5^, 10^6^, or 10^7^ OBs/mL on 1 cm^2^ diameter fresh mulberry leaf. It took about 12 h for a silkworm to eat the entire virus-coated leaf. We noted the time at which the larvae had consumed the entire leaf piece after 12 h of feeding as 0 h. Silkworm larvae of the third instar, fed with fresh mulberry leaves treated with distilled water, served as a negative control group. Larvae were then reared on fresh mulberry leaves, and larvae from all experimental groups were maintained under the same conditions. Mortality was recorded daily for 10 days. Each group contained 72 larvae.

### 2.6. Mutagenesis Analysis of the Viral Target Genes

Genomic DNA was extracted from eight whole larvae of the transgenic and wild type (WT) silkworms treated with 10^6^ OBs/larva using standard SDS lysis–phenol treatment. PCR amplification was performed with BmNPV-specific primers at 48 h post-infection (hpi) using 50 ng genomic DNA as the template. PCR products were cloned into the pJET-1.2 vector (Fermentas) and sequenced. Primer sequences are listed in Appendix A.

### 2.7. RNA Isolation and Quantitative Real-Time PCR (qPCR) of Viral Genes

Total RNA was extracted from whole larvae treated with 10^6^ OBs/larva using the Trizol reagent (Invitrogen), as per the manufacturer’s protocol. Whole larvae samples were separately collected from silkworms in the TG-A, TG-B, TG-C, and WT groups treated with 10^6^ OBs/larva (*n* = 6 per group) harvested for analysis every 12 h from 0 to 72 hpi. RNA was quantified by spectrophotometry and purity was evaluated by gel electrophoresis. First-strand cDNA was prepared with the Thermo RevertAid First Strand cDNA Synthesis Kit (Thermo Fisher Scientific, Waltham, MA, USA) following the manufacturer’s protocol. To quantify the relative transcription levels of targeted genes, gene-specific primer sets were designed and used for qPCR performed with 2× SYBR Green PCR Master Mix (Toyobo). The housekeeping gene *Rp49* (GenBank accession number AB048205.1) [28] was used as an internal control to standardize the variance of the different templates, and analyzed using the 2^−^^△△CT^ method [29]. The mRNA measurements were quantitated in three independent biological replicates and three independent technical replicates. Primers are listed in Appendix A.

### 2.8. Microscopy of Infected Midguts

The midguts of TG-A, TG-B, and TG-C transgenic silkworms and WT silkworms treated with 10^6^ OBs/larva were collected at 60 hpi on ice and fixed in Qurnah’s fixative (methanol/chloroform/acetic acid, 6:3:1 *v*/*v*/*v*) overnight. Samples were dehydrated in anhydrous ethanol, followed by xylene and embedded in paraffin. Paraffin-embedded midguts were sectioned with a Leica RM2235 microtome to obtain 5 μm thick sections. Sections were dewaxed using xylene, serially rehydrated with ethanol, and stained with a mixture of hematoxylin and eosin. Photographs were obtained by fluorescence microscopy (Olympus BX51).

### 2.9. Statistical Analysis

Each experiment was biologically repeated three times. The experimental data were analyzed using one-way ANOVA or two-way ANOVA, * *p* < 0.05, ** *p* < 0.01, *** *p* < 0.001. The values are shown as the mean ± standard deviation of three independent experiments.

## 3. Results

### 3.1. Generation of Transgenic Silkworms

We previously described a CRISPR/Cas9 anti-BmNPV system that targeted intermediate early genes [22]. Here, we targeted the essential genes *lef8* and *lef9* simultaneously at two sites per gene (Figure 1A). The *lef8* and the *lef9* genes from the BmNPV genome were cloned and the verified sequences were used as selected target sites for sgRNAs. The Cas9 protein and the fluorescent maker EGFP under the *IE1* promoter and the sgRNA arrays under the control of the *B. mori U6* promoter were expressed in the germline (Figure 1B), as previously described [22]. We injected the vector into the cytoplasm of zygotes and identified putative transgenic animals based on EGFP expression. Three independent G_0_ transgenic lines—TG-A, TG-B, and TG-C—were selected, and the genomic insertion sites were identified by inverse PCR and Sanger sequencing. The results showed that the transgenes in TG-A, TG-B, and TG-C line were located in chromosome 11, chromosome 10, and chromosome 16, respectively (Figure 1C–E).

### 3.2. Normal Growth and Development in the Transgenic Lines

In order to ensure that the growth and development of the transgenic lines were normal, we carefully compared the developmental time and weight in larval stages with the WT. In transgenic silkworms, we found no significant differences in body weight between the transgenic and WT lines, with approximately 1 g per larva in fifth larval instar day 4 (Figure 2A). In addition, we did not observe any differences in larval progression between WT and transgenic animals within the 17–19-day range (Figure 2B–E).

### 3.3. Transgenic Silkworms Showed Higher Survival Rate after BmNPV Infection

To determine whether transgenic silkworms are more able to survive viral infection than WT silkworms, the third instar larvae of the transgenic and WT lines were orally infected with BmNPV with 10^5^, 10^6^, and 10^7^ OBs/larva, and mortality was monitored from 0 to 10 days post-infection. The survival rates of the TG-A, TG-B, and TG-C were approximately 97.22%, 100%, and 100%, respectively, at 10 dpi with 10^5^ OBs/larva, whereas only 79.17% of WT silkworms were alive at 10 dpi (Figure 3A). Under 10^6^ OBs/larva infection condition, the survival rate of WT silkworms was 63.89%, which was significantly lower than that of transgenic silkworms (TG-A, TG-B, and TG-C were 93.06%, 88.89%, and 94.44%, respectively) (Figure 3B). In the experimental group infected with 10^7^ OBs/larva, all WT silkworms were dead by 7 dpi, whereas at 10 dpi, TG-A, TG-B, and TG-C exhibited 79.17%, 77.78%, and 73.61% of transgenic silkworms remaining alive, respectively (Figure 3C). WT silkworms infected with BmNPV had enhanced locomotor activity relative to infected transgenic animals, and at 60 hpi they became yellow and puffy with swelling of the segmental membrane (Figure 3D). In contrast, the TG silkworms appeared normal in shape, with no difference from healthy uninfected silkworms.

### 3.4. Targeted Mutations of BmNPV lef8 and lef9 in Infected Transgenic Silkworms

To evaluate gene-editing efficiencies and to precisely analyze effects on *lef8* and *lef9* in BmNPV-infected silkworms, DNA samples were extracted from whole larvae of transgenic and WT silkworms treated with 10^6^ OBs/larva of BmNPV. Sanger sequencing and qPCR analyses were performed. The three transgenic lines have the same antiviral efficiency (Figure 3) and the CRISPR system cuts the virus genome randomly; therefore, we then took TG-A as an example for detecting the knockout efficiency of the target gene by the transgenic system. In TG-A larvae infected with BmNPV, PCR analysis indicated various types of genome editing events presented in *lef8* and *lef9* (Figure 4A). qPCR analysis of RNA samples extracted from infected transgenic silkworm larvae and WT larvae at 48 hpi revealed that *lef8* and *lef9* mRNAs were expressed at negligible levels in the transgenic larvae, whereas levels of these mRNAs were higher in WT larvae (Figure 4C). No gene editing was found in the BmNPV DNA from WT, while there are many types of mutations, insertions, small and large deletions in the BmNPV DNA recovered from the transgenic silkworms (Figure 4D). These results indicated that the transgenic CRISPR/Cas9 system effectively disrupted BmNPV *lef8* and *lef9*.

### 3.5. Microscopy Analysis of Midgut Cells in the Infected Transgenic and WT Silkworms

Baculovirus infection starts in the larval midgut cells and then spreads to other tissues including fat body, hemocytes, trachea, neurons, and the brain [30]. The midgut is vital for nutrient metabolism and is a source of innate immunity in the silkworm. A micro-sectioning analysis showed that at 60 hpi the midgut columnar epithelial cells were normal, and that there was no structural damage in the midgut tissues of the infected transgenic silkworms, whereas midgut columnar epithelial cells were disordered in the infected WT silkworm (Figure 5).

### 3.6. BmNPV Replication was Inhibited in Transgenic Silkworms

Baculovirus infection of permissive insects proceeds in a cascade fashion with the transcription of immediate early, early, late, and very late genes. Transcription of early genes activates late-phase gene expression. To explore BmNPV gene expression in the transgenic silkworms, the levels of different viral transcripts, immediate early gene *ie-1*, early gene *p143,* late gene *vp39*, and very late gene *p10*, were quantitatively analyzed. qPCR data showed a clear decrease in the expression levels of each of these genes at 60 hpi with 10^6^ OBs in transgenic silkworms compared with WT animals (Figure 6A–D).

To further examine the effect of *lef8* and *lef9* deletion on virus replication in the transgenic animals, total DNA was extracted from silkworms treated with 10^6^ OBs/larva, and the relative copy numbers of several representative BmNPV genes were analyzed by qPCR in WT and transgenic silkworms. In the WT silkworms, relative copy numbers increased up to 72 hpi, whereas very little BmNPV DNA was detected in transgenic silkworms (Figure 6D,E). These results showed that BmNPV replication was significantly inhibited in the transgenic silkworm.

## 4. Discussion

BmNPV is highly pathogenic to *B. mori*, and infections from this virus result in considerable economic damage to the sericulture industry. Targeted genomic modification of BmNPV is a powerful antiviral strategy. The CRISPR-Cas9 genome-editing system has been used to knock out key BmNPV genes, including *ie-1* and *me53* [22], *ie-0* and *ie-2* [31], and *iap2* [32], and all resulting in the inhibition of viral replication. This study engineered transgenic silkworms that express Cas9 and sgRNAs, targeting the BmNPV essential genes *lef8* and *lef9*. This also resulted in the inhibition of viral replication; the transgenic silkworms had significantly increased resistance to BmNPV infection compared with WT silkworms. This result showed an antiviral effect similar to our previous study which reported using *ie-1* and *me53* as targets [22]. In our transgenic Cas9 system, somatic mutagenesis were induced in Cas9-expressed and sgRNA-expressed lines, and the cleavage efficiency may vary in different lines and individuals. Herein, the PCR and qPCR results indicated that the transgenic CRISPR/Cas9 system effectively disrupted BmNPV *lef8* and *lef9*.

The host provides energy and biosynthetic precursors necessary for baculovirus replication, and editing-related host genes could improve resistance to BmNPV. *Bmcas-1*, *Bmcytc*, *Bmapaf-1*, and *BmNc* are all expressed at significantly different levels in resistant strains [33]. *Bmcas-1* and *BmNc* function in the apoptosis pathway in vitro to influence the pathology of BmNPV [34,35]. These genes could potentially be manipulated as an antiviral strategy, but the antiviral strategies showed limited success because they were identified in BmN cell lines [34,35]. In addition, targeting host genes may have impacts on host developments. Therefore, results suggest that targeting viral genes may be more effective than targeting host genes for generating viral resistant strains.

In addition to BmNPV, viruses such as *B. mori cytoplasmic polyhedrosis virus*, *B. mori densovirus*, and *B. mori infectious flacherie virus* infect silkworms, leading to significant economic losses. Currently, there are few effective measures to mitigate the effects of viral diseases in silkworms. The approach we used here will be further explored so as to generate transgenic silkworms resistant to other viruses in the future.

## Figures and Tables

**Figure 1 viruses-14-01119-f001:**
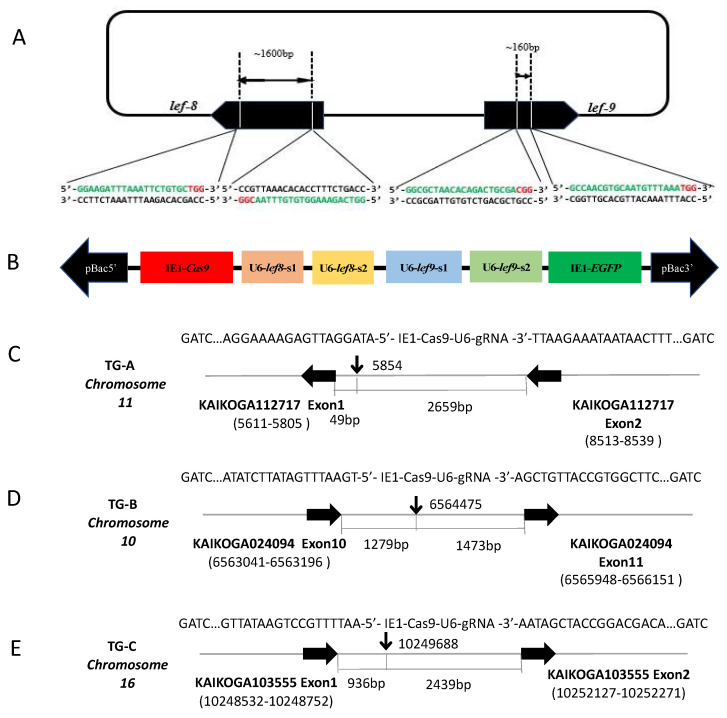
Construction of transgenic silkworm lines that express Cas9 and sgRNAs targeting BmNPV *lef8* and *lef9*. (**A**) Schematic representation of the location of targeted sequences within the BmNPV genome. The rectangle represents the BmNPV genome, and the two large black arrows represent the *lef8* and *lef9* genes. The drawing is not to scale. The target sequences are green, and the PAM sequences are red. (**B**) Schematics of the plasmid *pBac* [*IE1*-EGFP-*IE1*-Cas9-U6-*lef8* + *lef9* sgRNAs] for expression of Cas9 and sgRNAs. (**C**–**E**) Locations of genomic insertions of the *pBac* [IE1-EGFP-IE1-Cas9-U6-*lef8* + *lef9* sgRNAs] construct in TG-A (**C**), TG-B (**D**), and TG-C (**E**) transgenic *B. mori* lines. The vertical arrows indicate the insertion sites in the chromosome. The horizontal black arrows represent contiguous genes of the insertion sites. The interspace distance between the insertion sites of the construct and its contiguous genes is indicated in base pairs.

**Figure 2 viruses-14-01119-f002:**
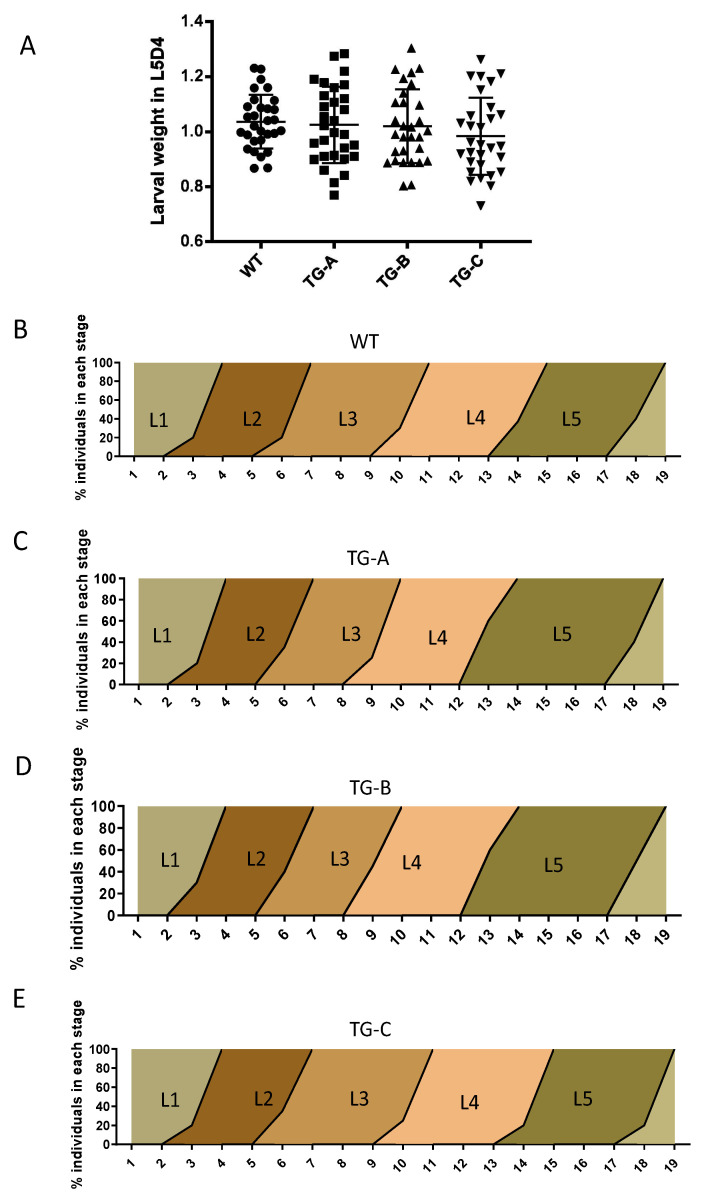
Normal growth and development in the transgenic lines. (**A**) No significant change in larval weight was observed between WT and transgenic animals. (**B**–**E**) No significant change in the stages of larval development in WT and TG-A, TG-B, and TG-C (L1, first instar; L2, second instar; L3, third instar; L4, fourth instar; L5, fifth instar).

**Figure 3 viruses-14-01119-f003:**
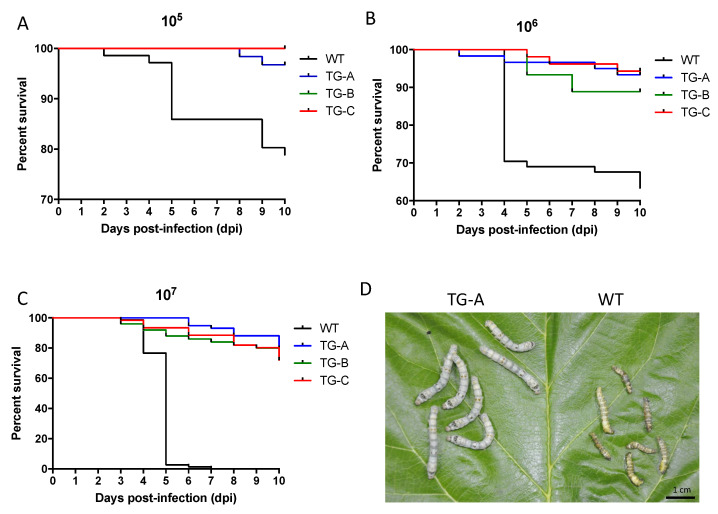
Transgenic silkworms had higher survival rates than wild type silkworms after infection with BmNPV. (**A**–**C**) Time course of survival rate of larvae after oral inoculation with 10^5^ (**A**), 10^6^ (**B**), and 10^7^ (**C**) OBs/larva. Each group included 72 larvae. The mortality was scored at 10 dpi. (**D**) Photographs of representative TG-A and wild type silkworms at 60 hpi with 10^6^ OBs/larva.

**Figure 4 viruses-14-01119-f004:**
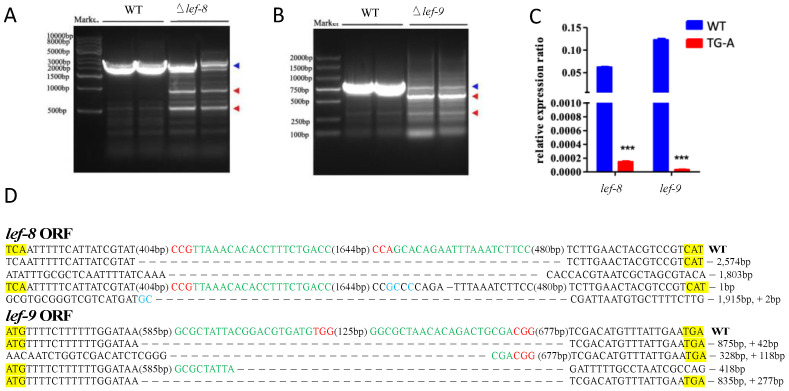
The BmNPV genome is edited in transgenic silkworms. (**A**,**B**) PCR-based amplification of regions in *lef8* (**A**) and *lef9* (**B**) in TG-A and WT silkworms infected with 10^6^ OBs/larva. The blue arrowheads indicate the complete full-length gene sequence and red arrowheads indicate the truncated edited gene sequence. (**C**) The *lef8* and *lef9* mRNA levels in TG-A and WT silkworms infected with 10^6^ OBs/larva. mRNA levels were normalized to those of *Bmrp49*. (**D**) Various fragment deletions were detected in TG-A silkworms. The red sequence indicates the PAM sequence. For each gene, the open reading frame in the WT sequence is shown at the top, with the start and stop codons highlighted in yellow, the target sites in green, the PAM sequences in red, and the inserted bases in blue. The net change in length caused by each indel mutation is given to the right of the sequence (+, insertion; −, deletion).

**Figure 5 viruses-14-01119-f005:**
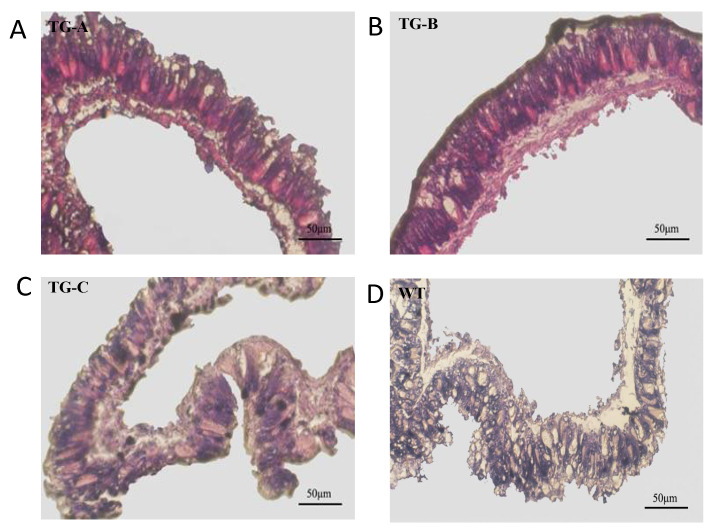
Microscopy of midguts of the transgenic and WT silkworms after BmNPV infection. The midguts of TG-A (**A**), TG-B (**B**), TG-C (**C**), and WT (**D**) silkworms infected with 10^6^ OBs/larva at 60 hpi were collected and subjected to HE staining microscopy. The midgut cells of the infected transgenic silkworms were tidily arranged; however, those of the infected WT silkworms were dislodged.

**Figure 6 viruses-14-01119-f006:**
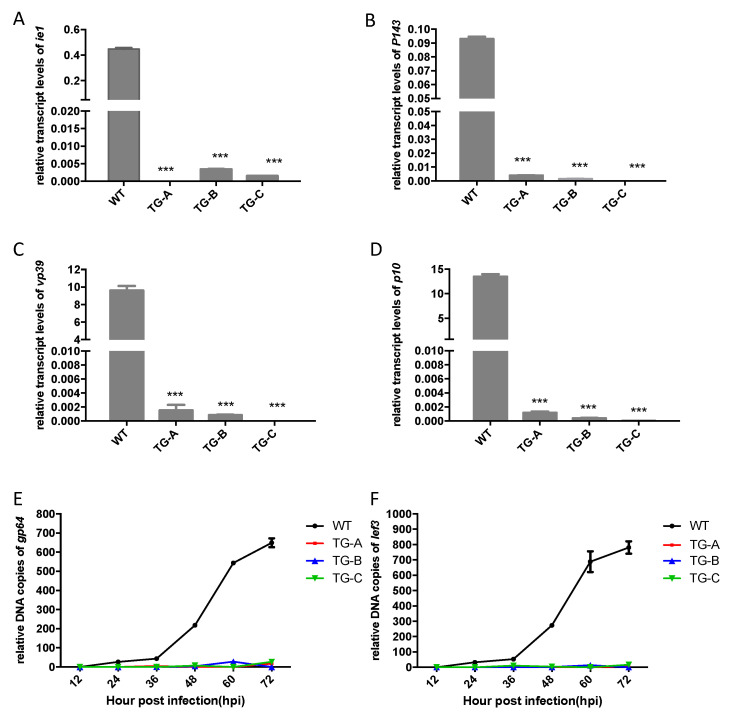
NPV transcription and replication are inhibited in transgenic silkworms. (**A**–**D**) Expression levels of *ie1* (**A**), *p143* (**B**), *vp39* (**C**), and *p10* (**D**) transcripts at 60 hpi in the transgenic and WT silkworms infected with 10^6^ OBs/larva of BmNPV. Transcript levels were determined relative to levels of *BmRp49*. Data are means ± SEM. *** *p* < 0.001 by one-way ANOVA. (**E**,**F**) Relative numbers of copies of viral DNA from the regions of *gp64* (**E**) and *lef3* (**F**) in the transgenic and WT silkworms at different points after viral inoculation with 10^6^ OBs/larva.

## Data Availability

Not applicable.

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
