# Peer review of "CRISPR/Cas9-Mediated Disruption of the lef8 and lef9 to Inhibit Nucleopolyhedrovirus Replication in Silkworms"

_viruses, 2022, doi:10.3390/v14061119_

Round 1

Reviewer 1 Report

In this work, liu et al utilized CRISPR/Cas9 system to disrupt BmNPV lef8 and lef9 genes and showed the successful resistance to BmNPV infection in silkworms. It is an interesting and meaningful research. The data presented are generally strong, and appear convincing. The work is based on a very strong foundation with clear and robust data, but would benefit with further experiments to help strengthen the main conclusions and to better understand.

Major Comments

  1. In Fig4, the led-8 and lef-9 genes were detected, but lack of the protein levels of lef-8 and lef-9.
  2. Could these transgenic silk worms completely the infected virus? Using this CRISPR-Cas9 system to generate the silkworm for resisting the infection of BmNPV, how about the possibility of the appearance of a mutant strain of BmNPV? If these transgenic silkworms raised together with the wild type with BmNPV infection for a long time, how about the result?
  3. In discussion part, the author needs to strengthen, including the advantages and disadvantages of using engineered transgenic silkworm by CRISPR/Cas9 strategy.
  4. All microscopy analysis should be described in specific pathologic terminology, such as “columnar epithelium”. Also, there was obvious difference between TG-C and TG-A/B, such difference should be specifically described but not mentioned.

Minor Comments

  1. Line66, “showed enhanced the antiviral efficacy” should be “showed the enhanced antiviral efficacy”
  2. Line76, “The multivoltine, non-diapausing silkworm strain B. mori Nistari (ref)” lack of Reference.
  3. Line195, “which is significantly lower than that” should be “which was significantly lower than that”
  4. Line202, “in shape and no different from healthy silkworm without infection.” Should be “difference”
  5. Line204, “Transgenic silkworms have higher survival rates than wild-type” should be “had”
  6. Line213, “so we then” should be “we then”, delete “so”, as there was “because” in the former sentence
  7. Line202, “while there are many types of mutations, insertion” should be “while there were many types of mutations, insertions”
  8. Line250, “To explore BmNPV gene express in” should be “expression”
  9. Line278, “previous study reported which using” should be “previous study which reported using”
  10. bLine285, “ut the antiviral strategies are with limited success because” should be “but the antiviral strategies showed limited success because”

Author Response

Response to Reviewer 1 Comments

In this work, liu et al utilized CRISPR/Cas9 system to disrupt BmNPV lef8 and lef9 genes and showed the successful resistance to BmNPV infection in silkworms. It is an interesting and meaningful research. The data presented are generally strong, and appear convincing. The work is based on a very strong foundation with clear and robust data, but would benefit with further experiments to help strengthen the main conclusions and to better understand.

Response: Thanks for your positive evaluation. We carefully read all the comments and revised the whole manuscript.

Major Comments

Point 1: In Fig4, the led-8 and lef-9 genes were detected, but lack of the protein levels of lef-8 and lef-9.

Response 1: Thanks for your comments. We don't have antibodies to the lef-8 and lef-9 gene to test the protein levels. But we detected the mRNA of lef-8 and lef-9 gene by qPCR and RT-PCR (Figure 4). The lef-8 and lef-9 in the TGs had been sequenced. There are many types of mutations, insertions, small and large deletions in the BmNPV DNA ((Figure 4D). These results indicated that the transgenic CRISPR/Cas9 system effectively disrupted BmNPV lef8 and lef9.

Point 2: Could these transgenic silk worms completely the infected virus? Using this CRISPR-Cas9 system to generate the silkworm for resisting the infection of BmNPV, how about the possibility of the appearance of a mutant strain of BmNPV? If these transgenic silkworms raised together with the wild type with BmNPV infection for a long time, how about the result?

Response 2: Thanks for your comments. There was very little BmNPV DNA was detected in transgenic silkworms treated with 106 OBs/larva (Figure6 E and F). These results showed that BmNPV replication was significantly inhibited in the transgenic silkworm with 106 OBs/larva. But in the experimental group infected with 107 OBs/larva, approximately 20% of transgenic silkworms dead. Although the virus titer is lower than 106 OBs/larva in the practical rearing farm, we are trying very hard to improve the antivirus system to achieve higher level of resistance.

There are many types of mutations, insertions, small and large deletions in the BmNPV DNA (Figure 4D). Because lef-8 and lef-9 gene is the key gene of virus replication and proliferation, mutation will cause the virus to not be amplified, so the probability of obtaining the mutant virus with normal amplification is relatively low.

At present, we are rearing transgenic silkworm independently with wild type silkworms, the death rate of wild type silkworm is much higher than transgenic strains. If these transgenic silkworms raised together with the wild type with BmNPV infection for a long time, we think the transgenic lines will survive, the wild type silkworm will die. Because the hybridization will not happen (all the silkworm individuals will be harvest before pupation), the gene transfer will be impossible.

Point 3:  In discussion part, the author needs to strengthen, including the advantages and disadvantages of using engineered transgenic silkworm by CRISPR/Cas9 strategy.

Response 3: Thank you very much for pointing out our writing shortage. As suggested, we have strengthened the advantages and disadvantages of using engineered transgenic silkworm by CRISPR/Cas9 strategy in the discussion section (Line 295 - 298). The revised sentence is read as “In our transgenic Cas9 system, somatic mutagenesis be induced in Cas9-expressed and sgRNA-expressed lines, and the cleavage efficiency may vary in different lines and in-dividual. Herein, the PCR and qPCR results indicated that the transgenic CRISPR/Cas9 system effectively disrupted BmNPV lef8 and lef9.”.

Point 4: All microscopy analysis should be described in specific pathologic terminology, such as “columnar epithelium”. Also, there was obvious difference between TG-C and TG-A/B, such difference should be specifically described but not mentioned.

Response 4: Thanks for suggestion. We have corrected the sentence to make it clear in the revised version as suggested (Line 251 - 255). The revised sentence is read as “A micro-sectioning analysis showed that at 60 h p.i. the midgut columnar epithelial cells were normal and there was no structural damage in the midgut tissues in the infected transgenic silkworms, whereas midgut columnar epithelial cells were disordered in the infected WT silkworm (Figure 5).”.

Minor Comments

Point 1: Line66, “showed enhanced the antiviral efficacy” should be “showed the enhanced antiviral efficacy”.

Response 1: Revised as suggested (Line 66). The revised sentence is read as “the results showed the enhanced antiviral efficacy.”.

Point 2:  Line76, “The multivoltine, non-diapausing silkworm strain B. mori Nistari (ref)” lack of Reference.

Response 2: Revised as suggested (Line 76). We have corrected this error in the revised manuscript.

Point 3: Line195, “which is significantly lower than that” should be “which was significantly lower than that”

Response 3: Revised as suggested (Line 201). The revised sentence is read as “which was significantly lower than that of transgenic silkworms“.

Point 4: Line202, “in shape and no different from healthy silkworm without infection.” Should be “difference”

Response 4: Revised as suggested (Line 210). The revised sentence is read as “But the TG silkworms showed normal in shape and no difference from healthy silkworm without infection.”.

Point 5: Line204, “Transgenic silkworms have higher survival rates than wild-type” should be “had”

Response 5: Revised as suggested (Line 213). The revised sentence is read as “Transgenic silkworms had higher survival rates than wild-type silkworms”.

Point 6: Line213, “so we then” should be “we then”, delete “so”, as there was “because” in the former sentence

Response 6: Revised as suggested (Line 222). The revised sentence is read as “Because the three transgenic lines have the same antiviral efficiency (Figure 3) and the CRISPR system cuts the virus genome randomly, we then took TG-A as an example to detect the knockout efficiency of the target gene by the transgenic system.”.

Point 7: Line222, “while there are many types of mutations, insertion” should be “while there were many types of mutations, insertions”

Response 7: Revised as suggested (Line 230). The revised sentence is read as “while there are many types of mutations, insertions, small and large deletions in the BmNPV DNA”.

Point 8: Line250, “To explore BmNPV gene express in” should be “expression”

Response 8: Revised as suggested (Line 266). The revised sentence is read as “To explore BmNPV gene expression in the transgenic silkworms”.

Point 9: Line278, “previous study reported which using” should be “previous study which reported using”

Response 9: Revised as suggested (Line 294). The revised sentence is read as “to our previous study which reported using the ie-1 and me53 as a target”.

Point 10: bLine285, “ut the antiviral strategies are with limited success because” should be “but the antiviral strategies showed limited success because”

Response 10: Thanks for your comments (Line 304). The revised sentence is read as “but the antiviral strategies showed limited success because they identified in BmN cell lines”.

Reviewer 2 Report

A baculovirus disease-resistant transgenic silkworm developed using CRISPR/Cas technology is described in this manuscript. It provides a new idea for the development of new anti-virus varieties in sericulture. Some content in the text needs to be revised and supplemented:

1 In line 120, ‘…at 48 post infection…’ should be ‘…at 48 h post infection…’.

2 In line 192, ‘…0 to 10 days dpi.’ should be ‘…0 to 10 days post infection.’

3 There is no reference 35 in this article.

4 In figure 1B, there are two ‘U6-lef9-s1’. Is the second one ‘U6-lef9-s2’?

5 In figure 3, the percent survival of negative control group (treated with distilled water) needed be added as A or in figure legend. And what is the weight of surviving larvae in each group. These data could be added in supplement files.

6 Is there any difference in the baculovirus load in the dead larvae of the transgenic silkworm groups compared with the WT silkworm group?

Author Response

Response to Reviewer 2 Comments

Comments and Suggestions for Authors

A baculovirus disease-resistant transgenic silkworm developed using CRISPR/Cas technology is described in this manuscript. It provides a new idea for the development of new anti-virus varieties in sericulture. Some content in the text needs to be revised and supplemented:

Response: Thanks for the evaluation. We have eliminated some minor errors in the revised manuscript.

Point 1: In line 120, ‘…at 48 post infection…’ should be ‘…at 48 h post infection…’.

Response 1: Thanks for your comments. We have corrected this error in the revised manuscript (Line 122). The revised sentence is read as “PCR amplification was performed with BmNPV-specific primers at 48 h post infection”.

Point 2: In line 192, ‘…0 to 10 days dpi.’ should be ‘…0 to 10 days post infection.’

Response 2: Thanks for your comments. We have revised it in the revised manuscript (Line 198). The revised sentence is read as “and mortality was monitored from 0 to 10 days post infection.”.

Point 3: There is no reference 35 in this article.

Response 3: Thanks for your comments. We have rechecked all references and deleted the error “35”.

Point 4: In figure 1B, there are two ‘U6-lef9-s1’. Is the second one ‘U6-lef9-s2’?

Response 4: Thanks for your comments. We have corrected this error in the Figure1B.

Point 5: In figure 3, the percent survival of negative control group (treated with distilled water) needed be added as A or in figure legend. And what is the weight of surviving larvae in each group. These data could be added in supplement files.

Response 5: Thanks for suggestion. We have added the detail information of the percent survival of transgenic silkworm and WT animals in figure 3. We don't have measured the detail weight of surviving larvae in each group. But we found no significant variations in body size between infected transgenic silkworm and uninfected transgenic silkworm.

Revised as suggested (Line 205 - 207). The revised sentence is read as “In the experimental group infected with 107 OBs/larva, all WT silkworms were dead by 7 dpi, whereas at 10 dpi, TG-A, TG-B, and TG-C were 79.17%, 77.78%, and 73.61% of transgenic silkworms remained alive (Figure 3C).”.

Point 6: Is there any difference in the baculovirus load in the dead larvae of the transgenic silkworm groups compared with the WT silkworm group?

Response 6: We don't have detected the detail baculovirus load in dead larvae. But we have detected the detail baculovirus load in alive animals. The detail information in the section 3.6.
